# The Effect of Extrusion and Heat Treatment on the Microstructure and Tensile Properties of 2024 Aluminum Alloy

**DOI:** 10.3390/ma15217566

**Published:** 2022-10-28

**Authors:** Qilun Li, Xiaobo Zhang, Lei Wang, Jisen Qiao

**Affiliations:** 1School of Materials Science and Engineering, Lanzhou University of Technology, Lanzhou 730050, China; 2State Key Laboratory of Advanced Processing and Recycling of Non-Ferrous Metals, Lanzhou University of Technology, Lanzhou 730050, China

**Keywords:** 2024 aluminum alloy, extrusion pipe, mechanical properties, microstructure

## Abstract

Hot extrusion forming is one of the best cost-effective processing methods to obtain high-strength aluminum alloys. In order to obtain high performance 2024 aluminum alloy for the aero and automobile industries, this research comprehensively uses heat treatment and reverse isothermal extrusion technology to prepare 2024 alloy. The effects of homogenization, extrusion and post-extrusion annealing treatment on the microstructure and mechanical properties of 2024 aluminum alloy were discussed in detail. The results indicate that the grain refinement of the extruded alloy material is significant. The coarse eutectic microstructure at the grain boundaries was refined, and these grains tended to be uniformly distributed after the annealing treatment. Extruded 2024 aluminum alloy material mainly has S (Al_2_CuMg) and Al_7_Cu_2_Fe second phases. The appearance of a large number of S phases led to a significant improvement in the properties of the alloy with an increase in tensile strength and elongation of 176% and 547%, respectively. In addition, EBSD analysis showed a significant meritocratic growth in the extrusion direction with the appearance of Copper {112} <111> rolling weaving, which led to process hardening and the strength improvement of the alloy.

## 1. Introduction

With the rapid development of the aerospace industry, Al-Mg-Cu series aluminum alloys as important high-strength structural materials have attracted considerable attention in research and application [1,2,3]. As a superior Al-Mg-Cu alloy, 2024 aluminum alloy is mainly composed of aluminum, copper, magnesium, iron, silicon, and manganese. It has high strength, good plasticity and toughness, better fatigue resistance and corrosion resistance, and excellent cold and hot processing performance [4,5,6,7,8,9]. Therefore, 2024 aluminum alloy is applied to manufactured significant products on the premise of satisfactory strength and fatigue resistance. Up to the present, 2024 high-strength aluminum alloy has been used to manufacture the skeleton structure, wing ribs, skin, and other various parts in large civil airliners, commercial airliners and military aircraft industry.

For the as-cast 2024 aluminum alloy, there were large size grains, serious macroscopic segregation, and coarse eutectic structures at the grain boundary, which resulted in the degradation of mechanical properties. Therefore, given the defects existing in the casting process of 2024, the performance of the 2024 aluminum alloy is eager to be improved by regulating the alloy composition, plastic deformation, homogenization treatment, solution aging treatment, and other means [10,11,12]. In recent years, the study of 2024 aluminum alloy mainly focuses on the effect of plastic deformation and heat treatment on the microstructure and properties of the alloy. The nano-size precipitates can be obtained in the alloy by combining solution and aging, which is the main reason for the high strength of the alloy [6,13,14,15,16,17]. The main function of plastic deformation is to refine the grain size, homogenize the microstructure and densify the structure of the alloy. Typically, S and θ precipitated phases during the heat treatment process, which bring about reduction or elimination the coarse eutectic structure produced in the casting process, and excellent performance of the aluminum alloy products [14,18,19,20,21]. The reported results show that the general precipitation sequence of the S-phase in Al-Cu-Mg alloy is as follows [22,23,24]:(1)SSSS → Cu/Mg clusters → GPB zones → S′ phases → S phases

In the cold rolled 2324 aluminum alloy there were fiber {001} <110>, Copper {112} <111>, Brass {011} <211>, S {123} <634>, Goss {011} <100>, Cube {001} <100>, and R {124} <211> textures. The strength of the brass and S textures contribute significantly to the strength enhancement of the alloy [25]. Chia-Wei Lin studied the effects of the rotary forging process, complete annealing and solution treatment on the microstructure and properties of 2024 aluminum alloy. The results showed that rotary forging increased the strength but decreased the plasticity of the alloy, while the subsequent solution treatment improved the work hardening ability and increased the elongation of the alloy [26]. Yunlei Wang et al. investigated the organization and its properties of multi-pass cold-rolled aluminum sheets during graded annealing. Their results revealed that annealing at high temperatures weakens the strength of the texture and thus affects its mechanical properties. In contrast, lower annealing temperatures enhance the volume fraction of the cubic textures [27]. This indicates that the annealing temperature has a great influence on the formation of the weave structure and its strength of the alloy.

Therefore, plastic deformation combined with appropriate heat treatment are conducive to the uniform microstructure and excellent mechanical properties of aluminum alloy. In current research, reverse isothermal extrusion plastic forming was proposed to prepare 2024 aluminum alloy pipes, and the extrusion pipe was annealed. The purpose is to improve the microstructure inhomogeneity and to eliminate inclusions by extrusion technology to regulate the characteristics and microstructure of precipitated strengthening phase in the alloy and achieve the improvement of strength and ductility. The effects of extrusion and annealing treatment on the microstructure and properties of 2024 aluminum alloy were discussed.

## 2. Materials and Methods

The experimental materials used in this study were commercial 2024 semi-continuous cast aluminum alloy ingots. 2024 aluminum alloy chemical composition is shown in Table 1. The SMS MEER-36MN reverse extruder (Düsseldorf, Germany)was used to extrude at 380 °C with an extrusion ratio of 4.2 and an extrusion speed of 1.0 mm/s. The ingot was extruded into a profile with the specification Φ202 × 53 mm (wall thickness is 53 mm). The flow chart of the experimental method and the schematic diagram of the extrusion process are shown in Figure 1.

The extrusion process and heat treatment parameter of 2024 aluminum alloy used in the experiment are shown in Table 2.

In the research, the changes of microstructure and tensile properties of 2024 alloy in as-cast, extrusion and extrusion+annealing states were investigated in detailed. Three types of tensile tests of different stress states were designed and carried out on the alloy materials, including Uniaxial smooth tension (UT), Notched tension (NT) and Double notched shearing tension (DNT). The three different types of samples were shown in Figure 2. Tensile tests were carried out at room temperature on a Shimadzu Autograph AGS-X machine with a constant strain rate of 5.6 × 10^−4^·s^−1^. Three groups of validation tests were carried out for each stress state.

After the tensile test, the fractured part was cut out and then the characteristics of the tensile fracture were observed by the Quanta FEG-450 scanning electron microscope (SEM, Boynton Beach, FL, USA), and the distribution characteristics and chemical composition of the secondary phase were analyzed by an energy dispersive spectrometer (EDS). The samples with the size of 15 mm × 10 mm × 5 mm were taken out by wire cutting of the materials in three different treatment states and then the samples were polished for the microstructure analysis. Then, Keller reagent (1%HF + 1.5%HCl + 2.5%HNO_3_ + 95%H_2_O) was used to corrode the surface of the sample for 6~7 s at 25 °C. The microstructure was observed under LSM_800 laser confocal microscope (Zeiss, Oberkochen, Germany). The physical phases of the 2024 alloy specimens were characterized by Bruker D8 X-ray diffractometer (XRD, Billerica, MA, USA). The experimental conditions were monochromatic Cu target radiation, acceleration voltage of 40 kV, scanning rate of 4°/min, and scanning range 2θ of 15°~85°. An Oxford-SYMMETRY electron backscatter diffraction detector (Abingdon, UK) fitted with a Zeiss GEMINI 500 scanning electron microscope was used to scan and analyze the microstructures of the alloy materials.

## 3. Results and Discussion

### 3.1. Microstructure Analysis

Figure 3 shows the grain structure of 2024 aluminum alloy in as-cast, extruded, and annealed states. For the microstructure characteristics of the as-cast homogeneous treatment, Figure 3a indicated that the matrix microstructure is coarse equiaxed α-Al grains, mainly composed of α solid solution, and the grain size is very uneven. There are few defects and large eutectic phases at the grain boundary, and the eutectic phase distribution at the grain boundary is not uniform. Figure 3b display the grain structure after extrusion. It can be seen that the grains are elongated along the extrusion direction, forming a long fibrous grain structure. The black impurity phase formed at the grain boundary tends to be homogeneous, but its size is still relatively large. In Figure 3c, the annealed grains tend to be homogeneous due to recovery and recrystallization, and the annealed grains still retain most of the fibrous characteristics and extend along the extrusion direction. Compared with that before annealing, the black impurity phases at grain boundaries are evenly distributed and the intergranular segregation is reduced.

Figure 4 shows the XRD patterns of 2024 aluminum alloy under different treatment states. The content of the second phase and its changes during the extrusion and annealing treatments were determined by contrasting the standard PDF cards. The extruded 2024 aluminum alloy is mainly composed of Al matrix, a large amount of S phases and a few Al_7_Cu_2_Fe phases. Studies have indicated that the θ(Al_2_Cu) phase and T(Al_20_Cu_2_Mn_3_) phase are also present in 2024 aluminum alloy [13]. Their diffraction peaks cannot be observed in Figure 4 mainly due to their small amount, which is not enough for their diffraction by X-rays. Only the diffraction peaks of the Al matrix can be observed in the homogenized alloy material. The diffraction peaks of the S-phase are clearly observed after extrusion, which indicates that the S-phase precipitates during the extrusion process and its amount increases after the annealing treatment. In contrast, the amount of Al_7_Cu_2_Fe phase is basically unchanged, which is due to the fact that Al_7_Cu_2_Fe phase is a refractory phase.

Figure 5 shows the EDS-mapping distribution of 2024 aluminum alloy in the states of homogenization treatment, extrusion and annealing treatment. Figure 5a shows the scanning element distribution diagram of the homogenization treatment alloy, and the result indicated that the massive insoluble phase AlCuFeMn is formed in the alloy. There is S phase precipitation and concentrated distribution, which is an important reason to affect the properties of the alloy. Importantly, it is observed that the coarse insoluble phase AlCuFeMn and S phase become small particles in the extruded alloy, and the distribution tends to be homogeneous (in Figure 5b). The contents of Fe and Mn elements decrease, and Cu and Mg elements dissolve into Al matrix during extrusion. After quenching, there are supersaturated solid solutions of Cu and Mg elements in the alloy, and some of them form S-phase secondary precipitation. These supersaturated solid solution and S phase play a vital role in solid solution strengthening and precipitation strengthening [22]. It is worth noting that, as shown in Figure 5c, the S-phase size is smaller and the distribution is more uniform in the alloy after annealing treatment. The content of Cu and Mg also decreases, which makes the plasticity of the alloy increase further, but the corresponding strength decreases.

### 3.2. EBSD Microstructure Analysis

In this study, to further investigate the microstructure features of the alloy material, grain structure maps were obtained using electron backscatter diffraction (EBSD). Figure 6 shows the inverse pole figures (IPF) maps of 2024 aluminum alloy after homogenization treatment, extrusion and annealing treatment, respectively. The grains of the alloy in the homogenization treatment state in Figure 6a exhibit equiaxed grains and are dominated by high-angle grain boundaries (HAGBs). The grain size showed small equiaxed grains or sub-grains interspersed with large grain boundaries. From the figure, it can be statistically obtained that the average grain size in the as-cast state is 99.8 μm, and the grain size after extrusion is reduced to 13.27 μm. The grain structure of the extruded 2024 aluminum alloy is shown in Figure 6b. It can be observed that the grain microstructure is typical banded grains along the extrusion direction, and there are a large number of twins or sub-grains in the grain boundaries of the large deformation microstructure. The large banded grains are dominated by high-angle grain boundaries, while low-angle grain boundaries (LAGBs) are predominant within and between the banded grains. The overall grain size was refined significantly, and its average grain size was 13.57 μm. The microstructure of the annealed extruded alloy is shown in Figure 6c. The grains tend to be equiaxed from the typical ribbon organization, but still show an overall banded shape, with low-angle grain boundaries predominating. Figure 7 provides a statistical plot of the grain boundary orientation angle of alloy 2024 under different treatment conditions. Figure 7 shows the statistics of grain boundary orientation angle for alloy 2024 under different treatment conditions. As in Figure 7a, only 25% of the LAGBSs are observed during the homogenization treatment. After extrusion the percentage of LAGBs reaches 82.7%. It is well established that a large fraction of low-angle grain boundaries can effectively impede dislocation movement. The accumulation of dislocations causes an increased strength of the material.

Figure 8 shows the recrystallization diagrams of 2024 aluminum alloy with various states. The content percentages of recrystallized, sub-grains and deformed tissues are shown in Figure 8d. For the homogenization treatment alloy, the grains are mainly homogenized equiaxed grains. Figure 8b shows the recrystallization diagram of the alloy after extrusion, in which it can be seen that the microstructure of the alloy after extrusion is dominated by the deformation structure, which accounts for 71.15%. A small amount of substructure is distributed inside the grains of the deformed matrix. The content of recrystallized characteristic is similar to that of substructure, which is uniformly distributed at the grain boundaries of the deformed grains. When the extruded 2024 aluminum alloy is annealed, it can be seen that the deformed tissue is still in the majority, but it is reduced compared with that free of annealing, while the content of recrystallized grains and substructure has increased significantly, 14.62% and 24.65%, respectively. This result indicates that the reversion and recrystallization occurred during the annealing treatment, making the grain structure tend to be equiaxed, thus affecting the mechanical properties of the alloy material.

The polar images of the 2024 alloy in the homogenized, extruded and annealed states are shown in Figure 9. As shown in Figure 9a, no significant meritocratic orientation was found in the homogenized treated 2024 alloy sample. In contrast, the presence of a certain weave in the extruded 2024 alloy is confirmed in Figure 9b. Significant meritocratic growth along the extrusion direction (ED) occurs in the extruded alloy, with the presence of Copper {112} <111> rolled texture. The texture index reached 15.03 and the texture content was as high as 63.7%. The presence of the rolling texture allows the alloy to be anisotropic in the extrusion direction [28]. This texture is usually typical of the extruded or rolled deformed organization of FCC alloys. The annealed 2024 extruded tubing has a significantly lower Copper texture, with the content reduced to 46.4%. This reduction in weave content results in less work hardening of the alloy in the extrusion direction, which led to a reduction in yield strength of the alloy material.

### 3.3. Mechanical Properties

In order to verify the deformation ability of 2024 alloy material under different stress conditions, three tensile tests with different stresses were used in this study. Tensile specimens were cut on the extruded alloy along the extrusion direction and tested in axial tension on a testing machine. Figure 10 is the macroscopic fracture feature images of the 2024 aluminum alloy tensile sample with the states of homogenization treatment, extrusion and annealing treatment. In Figure 10a, the fracture morphology of the homogenized treatment shows a flush feature with brittle fracture along the grain boundaries. The fracture of the specimen after extrusion and annealing treatment fractures at an angle of 45° to the extrusion direction, and the fracture is an uneven tearing pattern, which is a macroscopic manifestation of tough nest fracture. From Figure 10a–c, it can be seen that the post-fracture length of the smooth tensile specimens increases sequentially, indicating that the plasticity of the specimens increases accordingly during the tensile process, which is consistent with the experimentally obtained performance results. In addition, the deformation of notched tensile and double-notched shear specimens in macroscopic form is minor. Since they received stress concentration, they are more likely to fracture under tensile stress. This also has led to the fact that their elongation is very low.

The tensile properties are shown in Figure 11. It can be seen from the result that the tensile strength and elongation of the extruded 2024 aluminum alloy are greatly increased compared with that of the annealed alloy, while the yield strength of the extruded 2024 aluminum alloy is decreased compared with that of the annealed material. The yield strength of uniaxial smooth tensile is less than that of notched tensile, but its elongation is much higher than that of notched tensile. This is because the alloy material is not easy to deform due to the high triaxial stress during notched tension, which leads to the increase of yield stress under the same strain condition. At the same time, the alloy material will fracture in advance under higher triaxial stress, resulting in a smaller strain in notched tensile than in uniaxial smooth tensile. The elongation of 2024 aluminum alloy under shear stress is smaller than that under uniaxial smooth tension, indicating that the deformation ability of the material under shear is smaller than that under uniaxial tension. Figure 11d shows the comparison of mechanical properties of as-cast, extrusion and extrusion annealing in uniaxial smooth drawing process. It can be observed that the tensile strength and yield strength of the extrusion and annealing treatment are better than those of the as-cast alloy. The tensile strength and yield strength of the extruded alloy are the highest, which are increased by 131.1 MPa and 85.46 MPa compared with the as-cast and annealed alloys, respectively. The elongation of the alloy increased to 15.7% after annealing.

Figure 12 is the tensile fracture morphology of the alloy after casting homogenization treatment under different stress states. The fracture morphology is characterized by intergranular fracture and cleavage fracture, and no secondary precipitates are found in the fracture. The same phenomenon is observed in the fracture morphology of notched tensile and double-notched shear tensile. It also shows that the tensile fracture mechanism of the cast 2024 aluminum alloy treated with homogeneous fire is cleavage fracture under different stress states.

Figure 13 and Figure 14 are the tensile fracture feature morphology images after extrusion and annealing treatment, respectively. The fracture morphology of uniaxial smooth tensile alloy was observed. The tensile fracture morphology of the extruded and annealed alloy was typical of ductile fracture, and there were large numbers of dimples around the tearing edge. The difference is that the dimple after extrusion is deeper, and S phase is precipitated at the bottom of the dimple, and the size of the precipitated phase is larger. The dimple of the annealed fracture was shallow and the precipitate size at the bottom was fine granular. By comparing their notched tensile morphology (in Figure 13b and Figure 14b), there are large amounts of dimples in the tensile fracture of the extruded alloy. The plane of the dimples is perpendicular to the plane of the tensile stress, and the direction of the dimples is the same as that of the tensile stress. The dimples in the annealed tensile fracture appear very shallow and there are some cleavage stages. The plastic slip patterns appear on the fracture of double-notched shear tensile, forming a large number of shear bands. Under the action of shear stress, the microcavity is elongated and an elliptical dimple is formed on the fracture, which is a shear dimple [29].

Notably, second phase particles in dimples cause dislocation accumulation. When the stress is high enough, the micropores expand and fracture occurs. The second phase particles in the extruded alloy are coarse, which can easily lead to fracture. This is the main reason why the plasticity of the extruded alloy is lower than that of annealed alloy.

Extrusion deformation has an influential effect on dislocation strengthening. Dislocations stimulate the precipitation of the second phase, while precipitation inhibits the movement of dislocations, and they interact with each other but co-exist. The accumulation of dislocations leads to stress concentration, which increases the hardness of the alloy [30,31]. In addition, the presence of some high-angle grain boundaries makes the alloy have a strong ability to regulate plastic deformation on the grain boundaries, and as a result the plasticity of the alloy improves [25].The rolling texture generated during extrusion is mainly distributed along the extrusion direction, so the alloy exhibits excellent mechanical properties in the extrusion direction [32]. The strength of the alloy is highest after extrusion, because the extrusion treatment increases the dislocation density and facilitates fine precipitation phases formation. After annealing treatment, the strength of the alloy decreases. That can be attributed mainly to the weakening of the Copper {112} <111> orientation during the annealing treatment, which leads to the reduction of grains in the <111>//ED direction and reduces the strength of the alloy [25,33]. The investigation presented by Yunlei Wang et al. indicated that high temperature annealing can weaken the weave strength due to the annealing softening of the alloy tissue after recrystallization [27].

## 4. Conclusions

As a common method to improve the properties of alloys, extrusion is widely used in the preparation of aluminum alloys. In this study, 2024 aluminum profiles with good properties were prepared by reverse extrusion. The effects of homogenization, extrusion and annealing treatment on the microstructure and tensile properties of 2024 aluminum alloy were discussed. Based on this study, it provides a research basis for the development of high performance aluminum alloys for aerospace and automobile power applications.

According to the experimental results, the following conclusions were drawn:(1)The tensile properties of 2024 aluminum alloy after homogenization and extrusion are excellent and the grain size is reduced. Due to homogenization and extrusion process, the grains tend to be homogeneous, and the insoluble phase of grain boundary is mostly dissolved in the matrix;(2)For the extruded 2024 aluminum alloy, the tensile properties are significantly improved, the ultimate tensile strength and yield strength are increased by 131.1 MPa and 60.07 MPa in comparison with the homogenization state, respectively, and the elongation is increased to 13.13%. After annealing, the elongation reaches 15.7%. More importantly, the similar effect of extrusion and annealing treatments on the mechanical properties was investigated with different tensile stresses;(3)There are supersaturated solid solutions and S precipitates of Cu and Mg elements in the extruded alloy, which form solid solution strengthening and precipitation strengthening and greatly enhance the strength of the alloy material;(4)The EBSD analysis showed that the extruded 2024 alloy had a clear meritocratic orientation along the extrusion direction, forming a typical Copper {112} <111> rolling texture. The annealed alloy undergoes reversion and recrystallization, and the strength of the Copper texture was reduced but still maintains a selective orientation in the extrusion direction.

## Figures and Tables

**Figure 1 materials-15-07566-f001:**
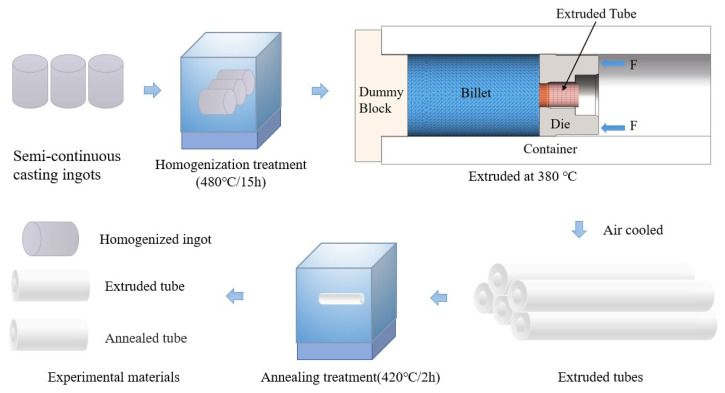
The flow chart of the experimental method and the schematic diagram of the extrusion process.

**Figure 2 materials-15-07566-f002:**
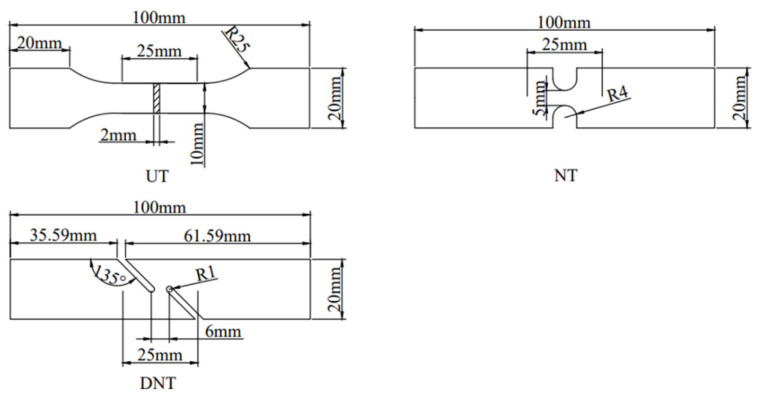
Dimensions of tensile samples (all in mm).

**Figure 3 materials-15-07566-f003:**
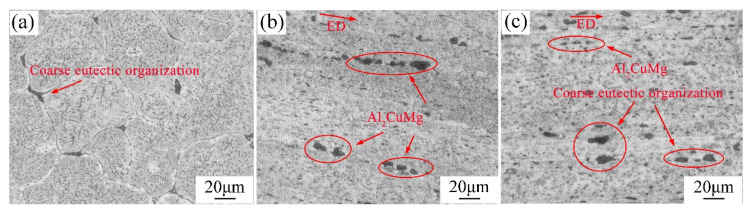
Microstructure diagrams of 2024 aluminum alloy with the states of (**a**) homogenization treatment, (**b**) extrusion, and (**c**) extrusion annealing treatment.

**Figure 4 materials-15-07566-f004:**
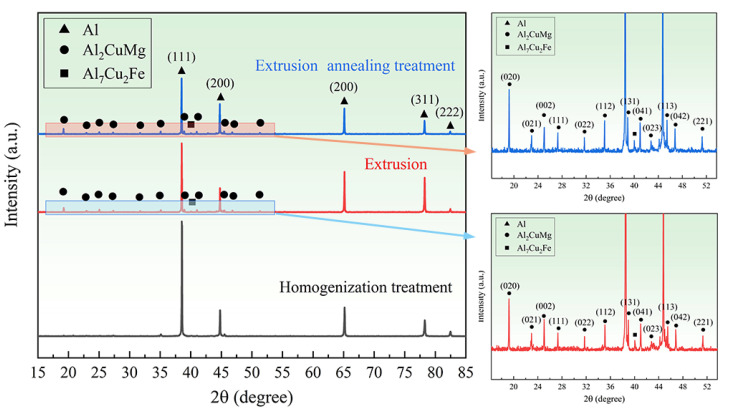
XRD patterns of 2024 aluminum alloy with the states of homogenization treatment, extrusion and extrusion annealing treatment.

**Figure 5 materials-15-07566-f005:**
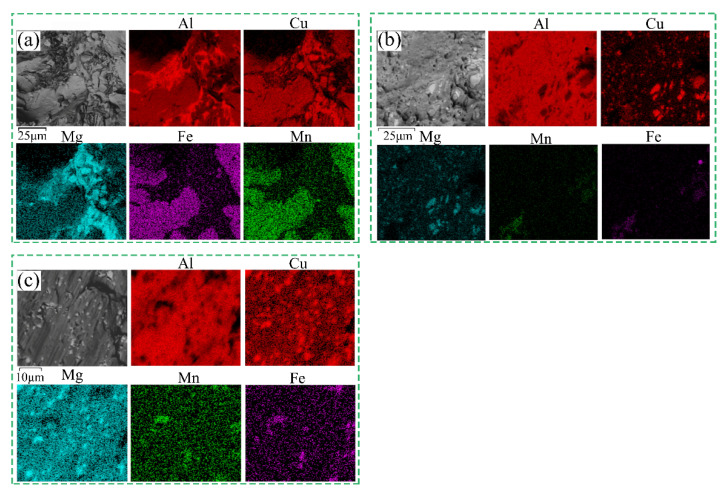
EDS element distribution maps of 2024 aluminum alloy in the states of (**a**) homogenization treatment, (**b**) extrusion and (**c**) extrusion annealing treatment.

**Figure 6 materials-15-07566-f006:**
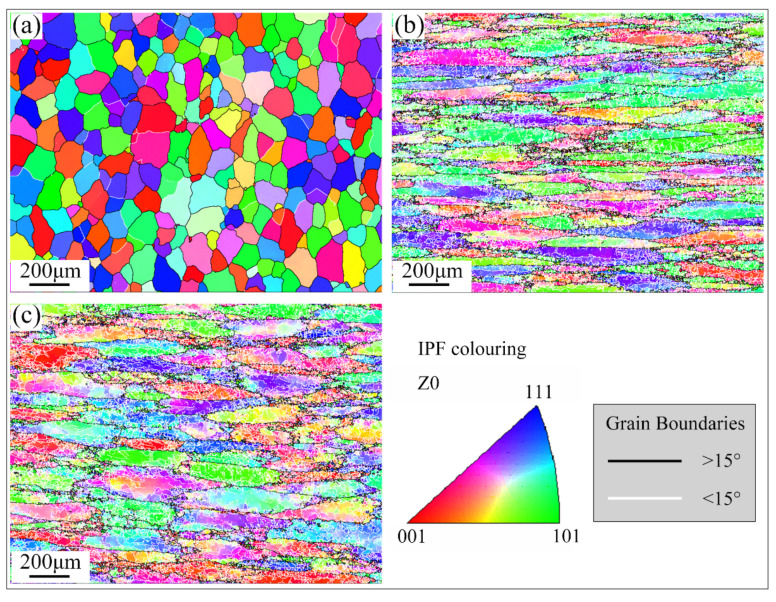
Inverse pole figures EBSD maps showing the grain structure of 2024 aluminum alloy by (**a**) homogenization treatment, (**b**) extrusion and (**c**) extrusion annealing treatment.

**Figure 7 materials-15-07566-f007:**
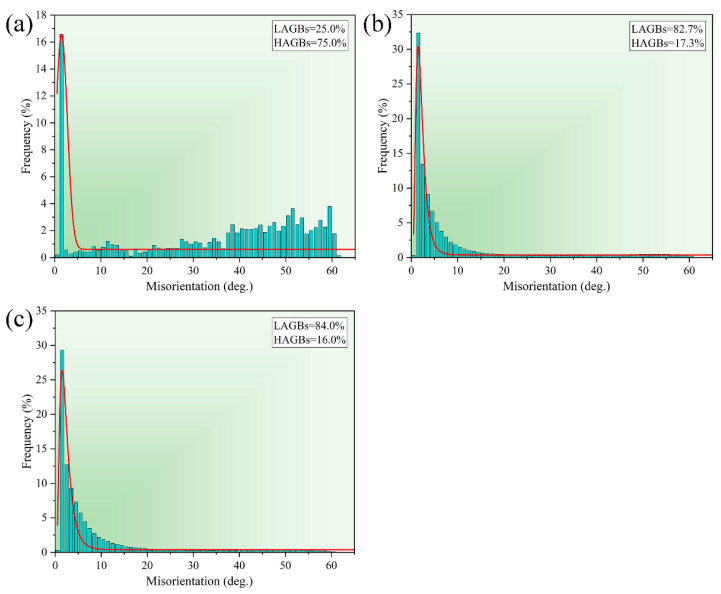
Histograms of grain boundary orientation angle for (**a**) homogenization treatment (**b**) extrusion treatment (**c**) extrusion annealing treatment of 2024 aluminum alloy.

**Figure 8 materials-15-07566-f008:**
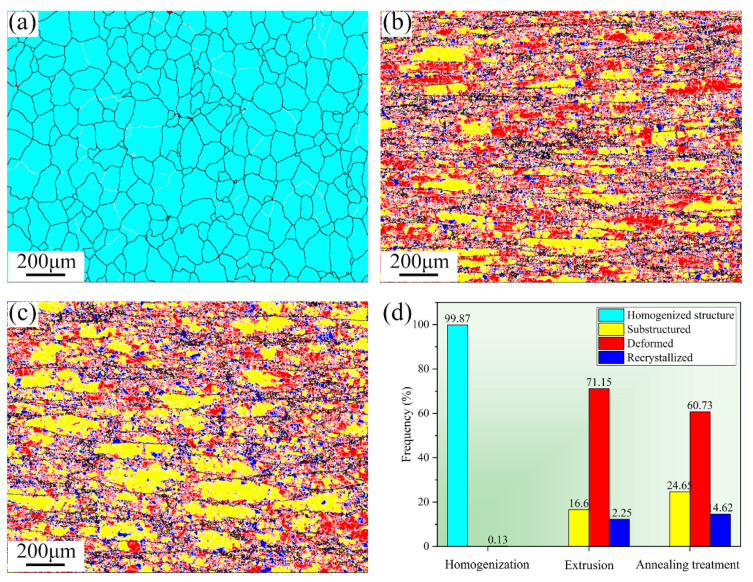
Recrystallization structure images of 2024 aluminum alloy by different treatments: (**a**) homogenization treatment, (**b**) extrusion, (**c**) extrusion annealing treatment, and (**d**) recrystallization content comparison.

**Figure 9 materials-15-07566-f009:**
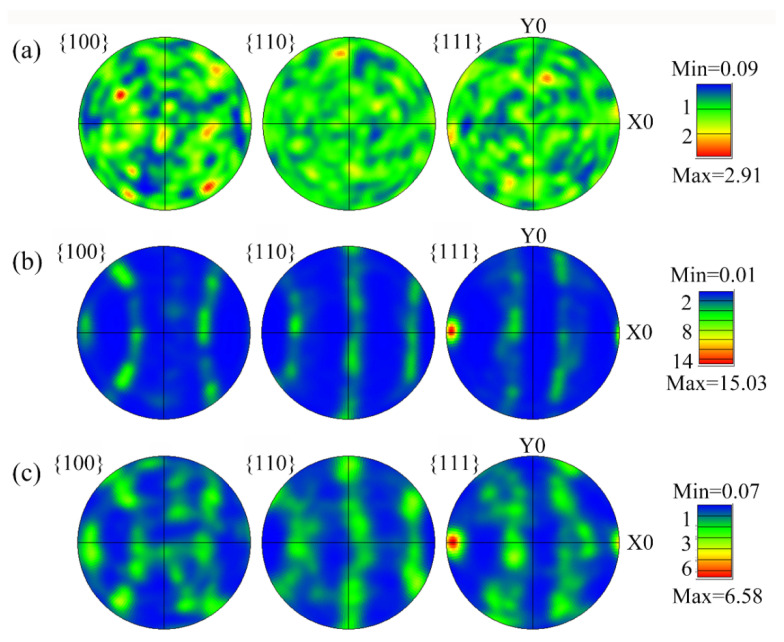
PFs of 2024 alloy with the states of (**a**) homogenization treatment, (**b**) extrusion, and (**c**) extrusion annealing treatment.

**Figure 10 materials-15-07566-f010:**
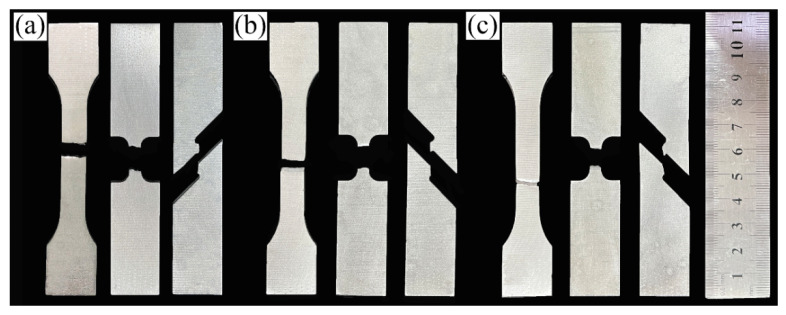
Macroscopic fracture feature images of tensile specimens with the treatment state of: (**a**) homogenization treatment, (**b**) extrusion and (**c**) extrusion annealing treatment.

**Figure 11 materials-15-07566-f011:**
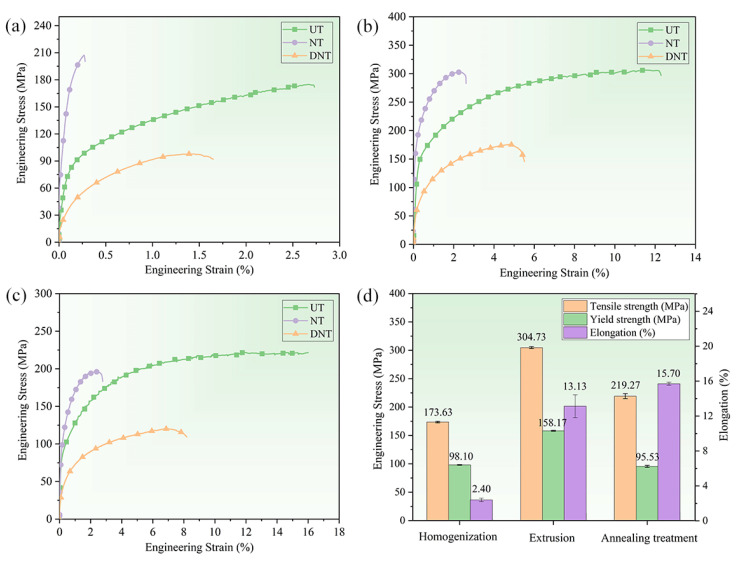
Engineering stress–strain curves of 2024 aluminum alloy under different stress conditions (UT, NT, DNT): (**a**) homogenization treatment, (**b**) extrusion, (**c**) extrusion annealing treatment and (**d**) comparison of uniaxial smooth tensile properties.

**Figure 12 materials-15-07566-f012:**
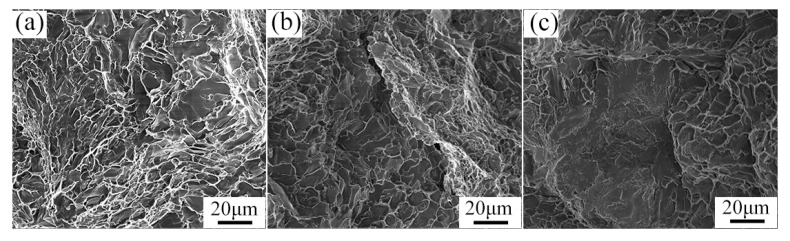
Fracture morphology images of 2024 aluminum alloy in the state of as-cast homogenization treatment (**a**) uniaxial smooth tensile, (**b**) notched tensile, and (**c**) double-notched shear tensile.

**Figure 13 materials-15-07566-f013:**
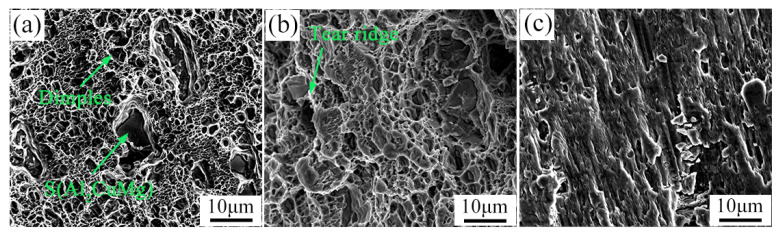
Fracture morphology images of 2024 aluminum alloy after extrusion (**a**) uniaxial smooth tensile, (**b**) notched tensile and (**c**) double-notched shear tensile.

**Figure 14 materials-15-07566-f014:**
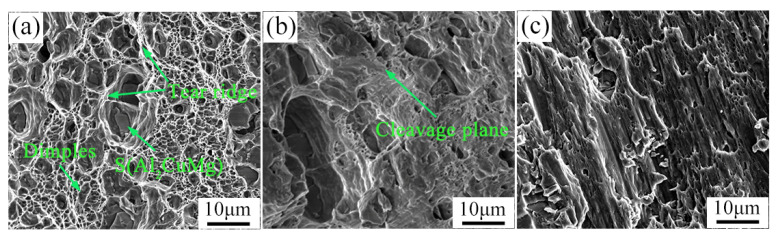
Fracture morphology images of 2024 aluminum alloy after annealing treatment (**a**) uniaxial smooth tensile, (**b**) notched tensile and (**c**) double-notched shear tensile.

**Table 1 materials-15-07566-t001:** Chemical composition of 2024 aluminum alloy.

Element	Cu	Mn	Mg	Si	Cr	Zn	Fe	Ti	Al
wt./%	3.8–4.9	0.3–0.9	1.2–1.8	0.5	0.10	0.25	0.5	0.15	Bal.

**Table 2 materials-15-07566-t002:** Treatment status of 2024 aluminum alloy.

Processing and Treatment	Treatment Process
Homogenization treatment	480 °C/15 h
Extrusion	380 °C
Annealing treatment	420 °C/2 h

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
