# Peer review of "The Effect of Extrusion and Heat Treatment on the Microstructure and Tensile Properties of 2024 Aluminum Alloy"

_materials, 2022, doi:10.3390/ma15217566_

Round 1
Reviewer 1 Report
Dear Authors;
In my opinion, the manuscript entitled “Study on tensile properties and microstructure of indirect extruded 2024 alloy” interesting results of studies. The work is not innovative.
I suggest that the authors should pay much attention to revise the manuscript by taking following points into account:
- The authors should more clearly emphasis the novelty of their work in the abstract and introduction.
- Materials and methods: authors need to provide the raw materials supplier.
- The authors should provide the characteristics and names of the apparatus (SEM, XRD, EBSD).
- The authors should provide more information about extrusion (input parameters, calibration,…).
- The authors should add new references in the introduction.
- A comparison with the existing published literature and discussion of the results are missing.
- Figure 2: to make a comparative study, all the figures must have the same magnification. I recommend that authors use a single magnification. Lack of information were observed in the pictorial representation.
- The interpretation of Figure 3 is poor.
- Figures 10, 11, and 12: I recommend that authors use a single magnification.
- Authors need to pay attention in Reference section also for formatting.
Sincerely Yours,
Reviewer
Reviewer 2 Report
The authors studied the tensile properties and microstructure of indirect extruded 2024 alloy. The manuscript had an interesting topic and was well-written, however it could only be accepted with the following minor revisions:
1. Please correct the grammar in the abstract and add a short summary of the study's findings.
2. Could you add the relevant references from the most recent year as about 50% of the references were out of date (over five years past)?
3. Instruction section can be improved and the research gap was not clearly described in the Introduction Section.
4. Please provide the ASTM code for UT, NT and DNT tests.
5. Figure 1 was not found the dimension’s unit.
6. Where did the author get the 2024 aluminum alloy? The authentication could be questioned.
7. The writing of the method was poor, and the various machines used in the experiment were not specifically explained in terms of type, model, origin, etc. for the tensile, SEM, XRD, etc.
8. Flow chart of the method should be included in the paper.
9. Figure 9(d), missing y-axis title and value.
10. Results and discussion was poor and lacks supporting reason with supporting literature for the obtained results and analysis (please improve).
11. It is suggested to add the limitation, future study, and develop implications for researchers in the conclusion section.
Reviewer 3 Report
Title: “Study on tensile properties and microstructure of indirect extruded 2024 alloy”
Manuscript Number: materials-1986103
Dear Editor,
I am attaching my review comments of the manuscript on a paper entitled “Study on tensile properties and microstructure of indirect extruded 2024 alloy”.
.
In this paper, the authors have studied the influence of homogenization, hot indirect extrusion, and annealing on microstructure evolution, tensile properties, and fracture surface morphology of Al2024 alloy tubes. The microstructure observations and fracture surface morphology were performed using optical, EBSD, and SEM microscopes. Moreover, the tensile test for different types of samples was also carried out using different tensile test samples shapes. The results of the paper were well presented and discussed. It is exciting research; the reviewer suggests accepting this paper for publication in the materials after a major revision to cover the following comments.
1- I think the paper's title must be correct to include the real content, like "the effect of extrusion of Al2024 tubes and heat treatment on their tensile properties".
2- Authors must provide systematic sketches or photos of the extrusion process. Moreover, the extrusion ratio and shape of the sample before and after extrusion must be added.
3- The experimental need more details. Did the annealing perform after extrusion? If so, the third case must be termed as extrusion + annealing
4- Indicate the position of the microstructure observations and the direction of the tensile test axis relative to the extrusion direction.
5- The EBSD confidence index (CI) used must be mentioned to judge the quality of the results.
6- Why were the grain and precipitate sizes not obtained for the microstructure observation? The increase in the tensile strength can be related to the discussion.
7- The results of the EBSD especially, those of the grain boundary maps and grain angle must be indicated with lines and use the following boundaries. (HAGBs misorientation angle >15 deg and low angle grain boundaries (LAGBs: misorientation angle<15 deg).
8- The value of the average grain boundary misorientation angle and the percent of the (HAGBs) (LAGBs) must be provided; both obviously affect the strengthen and the ductility of samples.
9- Why did the true stress/ true strain curves not use, as it is more accurate? Furthermore, to compare the different cases' elongations, it is better to use the uniform elongation, not the failure one.
10- The fracture surface mode in Fig. 8 needs further explanation.
11- The fracture surface morphology results need to be supported by the effect of each processing case on the size of the dimples and their distribution across the fracture.

Reviewer 4 Report
Dear authors,
You did an interesting work but the presentation of it does not satisfy completely. My remarks are as follows:
Please
improve Abstract by adding the one or two sentences about the motivation for this work. Also, emphasize the importance of the studied problem as well as novelty.
How values of chemical composition given in Table 1 was obtained?
How did you choose parameters in Table 2?
Please specify temperature of Keller reagent and type of XRD radiation.
It is suggested that the phases detected by XRD analysis be marked on the photomicrographs (Figure 2).
Please describe how XRD phase identification was performed.
In Discussion, you should try to confront your obtained results with results from other researchers dealing with similar issue.
Best regards
Round 2
Reviewer 1 Report
Dear Authors;
Thanks for your work in revising your manuscript according to the indicated
comments. The revised paper is well improved.
Therefore, I hope that this revised manuscript is now accepatble for publication
in the journal of "Materials".
With my best Regards
Reviewer 3 Report
I Accept the paper in the current form